# Rail-5k: a Real-World Dataset for Rail Surface Defects Detection

**Zihao Zhang**
The Key Laboratory of Road and Traffic Engineering, Ministry of Education
Shanghai Key Laboratory of Rail Infrastructure Durability and System Safety
Tongji University, Shanghai
1831412@tongji.edu.cn

**Shaozuo Yu**
Department of Computer Science
Tongji University, Shanghai
yushaozuo@tongji.edu.cn

**Siwei Yang**
Department of Computer Science
Tongji University, Shanghai
swyang.ac@gmail.com

**Bingchen Zhao**
Department of Computer Science
Tongji University, Shanghai
zhaobc.gm@gmail.com

**Yu Zhou**[*]
The Key Laboratory of Road and Traffic Engineering, Ministry of Education
Shanghai Key Laboratory of Rail Infrastructure Durability and System Safety
Tongji University, Shanghai
wqhuo2785@163.com

## Abstract

This paper presents the Rail-5k dataset for benchmarking the performance of visual algorithms in a real world application scenario, namely the rail surface defects detection task. We collected over 5k high quality images from railways across China, and annotated 1100 images with the help from railway experts to identify the most common 13 types of rail defects. The dataset can be used for two settings both with unique challenges, the first is the fully-supervised setting using the 1k+ labeled images for training, fine-grained nature and long-tailed distribution of defect classes makes it hard for visual algorithms to tackle. The second is the semi-supervised learning setting facilitated by the 4k unlabeled images, these 4k images are uncurated containing possible image corruptions and domain shift with the labeled images, which can not be easily tackle by previous semi-supervised learning methods. We believe our dataset could be a valuable benchmark for evaluating robustness and reliability of visual algorithms.

---

[*]Corresponding author

Submitted to the 35th Conference on Neural Information Processing Systems (NeurIPS 2021) Track on Datasets and Benchmarks. Do not distribute.

Table 1: Dataset compare

| Domain | Dataset | Task | # class | # image | # box per image | Resolution | Annotation Quality |
|---|---|---|---|---|---|---|---|
| Rail Defects | Delft [4] | cls | 6 | 3240 | 1 | $10^4$ gray-scale | image-level |
| | RSDDs [6] | seg | 2 | 195 | 5 | $10^4$ gray-scale | image-level |
| | CRRC [5] | det | 3 | >1000 | 1 | $10^4$ gray-scale | band-level |
| | Rail-5k(labeled) | det | 13 | 1100 | 22.9 | $10^7$ RGB | instance-level |
| Natural Image | VOC-2007 | det | 20 | 12974 | 3.1 | - | instance-level |
| | VOC-2012 [3] | det | 20 | 34071 | 2.7 | 469 x 387 RGB | instance-level |
| | ILSVRC-2014 [2] | det | 200 | 516840 | 1.1 | 482 x 415 RGB | instance-level |
| | MS COCO 2018 [16] | det | 80 | 163957 | 7.3 | - | instance-level |
| | OID V6 [14] | det | 600 | 1910098 | 8.4 | - | instance-level |

# 1 Introduction

The introduction of large scale annotated datasets such as ImageNet [2] greatly speeds up the development of deep-learning based vision algorithms [9]. Deep learning algorithms pre-trained on ImageNet [2] has also been shown to effectively transfer between domain and tasks such as object detection [21] or medical image analysis [12].

As an important basic infrastructure of human life, the maintenance and status analysis of railways has a real world economy and safety-focused value. However, current datasets in the railways domain are either limited in size  [6], quality of images [4, 5, 6], or the annotation types [5, 6] The limited size and quality of currently available dataset are not yet ready for support the training of deep learning methods.

Our dataset has enough high quality images captured from real-world railway to enable the training of deep learning models. Besides the labeled set with 1.1k images, we also provide a unlabeled set of 4k images to enable a semi-supervised setting. Several unique characteristics of our dataset also poses new challenges to vision algorithm. The first challenge is the long-tailed distribution of classes presented in out dataset, the imbalance ratio of the most majority class to the most minority class is up to 40.98, it has been shown that the long-tailed distribution would greatly hurt the performance of the learned model [17, 8].  Besides the long-tailed class distribution in the labeled set of the dataset, the unlabeled set of images also poses a difficult scenario of semi-supervised defect detection, semi-supervised object detection is a relatively new task with few recent works [7, 22], the previous method often assumes that the unlabeled set is also curated. However, in our case, the unlabeled set is uncurated with multiple unknown image corruptions and unseen object in the labeled set. Given these unique properties, we believe that our proposed dataset could not only facilitate the development of algorithms for rail surface defects detection, but also the development for a more robust vision model to handle the long-tailed distribution and possible corruptions in the unlabeled set.

# 2 Related work

Traditional inspection methods like subjective manual observation, sampling checking, are all qualitative or compensating methods, can not provide a digital and automatic decision-making basis for intelligent maintenance of the whole line. Our dataset mainly focus on the task of defects detection, we summarize relevant literatures in the section.

## 2.1 Natural Image Dataset

The surface defect detection tasks are most related to the tasks of object detection in visual algorithms. Common benchmarks for visual object detection are constructed using natural images such as Pascal VOC [3] and MS-COCO [16]. These dataset are mostly balanced in terms of class distributions. The LVIS [8] dataset proposed a larger collection of images with a long-tailed distribution of classes. Our proposed dataset also has a long-tailed distribution with respect to classes. Unlike the general natural image datasets, our dataset also presents fine-grained class definition due to the nature of railway images.

Table 2: Categories statistics.

| Class | Running surface | Contact band | Dark Contact Band | Spalling | Crack | Corrugation | Grinding |
|---|---|---|---|---|---|---|---|
| # Boxes | 1082 | 1093 | 773 | 12582 | 3785 | 3349 | 337 |
| #Images | 1080 | 1087 | 769 | 1005 | 375 | 445 | 179 |
| # Large | 1082 | 1092 | 773 | 1277 | 2965 | 3329 | 336 |
| #Medium | 0 | 0 | 0 | 5147 | 784 | 17 | 1 |
| # Small | 0 | 1 | 0 | 6148 | 36 | 3 | 0 |

| Class | Fastening | Spike Screw | Set Screw | Indentation | Burning | Welded Joint |
|---|---|---|---|---|---|---|
| # Boxes | 757 | 502 | 414 | 307 | 41 | 14 |
| # Images | 582 | 424 | 360 | 216 | 10 | 8 |
| # Large | 750 | 475 | 400 | 4 | 41 | 14 |
| # Medium | 7 | 27 | 14 | 237 | 0 | 0 |
| # Small | 0 | 0 | 0 | 66 | 0 | 0 |

## 2.2 Synthetic Corruption Dataset

There are also many datasets focusing on testing the robustness of deep-learning models under domain shift and image corruptions like ImageNet-C [10], CityScapes-C [18], and COCO-C [19]. However, the corruptions in these dataset are synthetic, generated using image processing techniques. Also, they are mainly used as the test set to test the robustness rather than the training set. In our dataset, the labeled dataset are well-curated, but the unlabeled set mat contains various real-world corruption, thus poses a new challenge for semi-supervised learning method.

## 2.3 Rail Defects Dataset

In the rail engineering domain, there are dataset focusing on the classification and detection of railway defects [23]. As for rail engineering, images are mostly in the form of atlas for manual reference. There are classification and detection [23] datasets of railway scene, as well as ultrasonic inspection datasets [11]. But still lacking of real-world datasets for rail surface defects. Faghih-Roohi *etal.* [4] collects and labels 100 x 50 resolution images in 6 defects classes. RSDDs datasets [6] contains 195 gray-scale images in 2 kinds of railway with segmentation mask. Feng *etal.* [5] collects thousands of images and annotate corruption, fatigue and spalling in band region. Datasets above are all collected by high-speed linear scan cameras with low resolution and coarse-grained annotation. As a consequence, they all fail to drive the training of real-world robust deep learning algorithms.

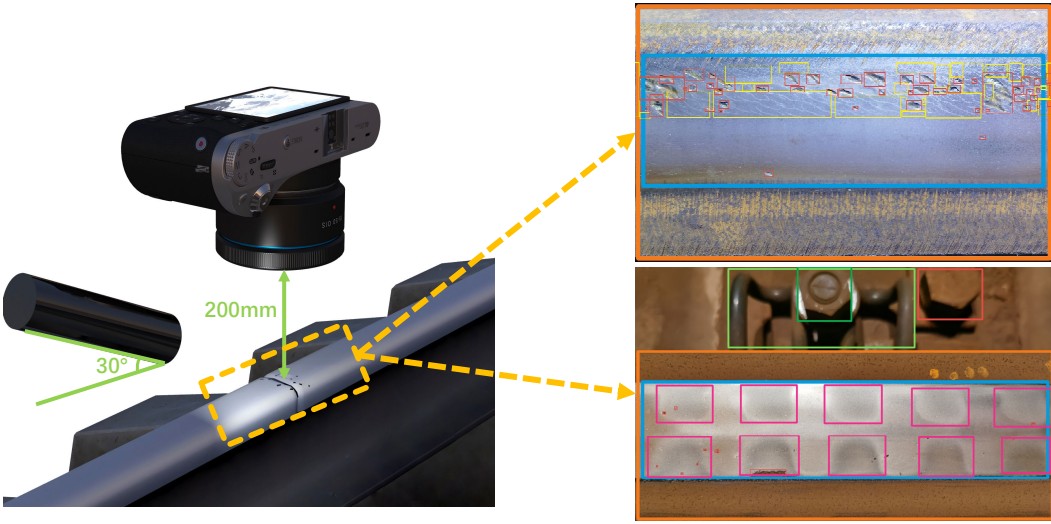

Figure 1: Typical image capture and annotations.

## 3 The Rail-5k dataset

### 3.1 Rail Image Acquisition

The rail surface defects are mostly caused by the metal fatigue under the constant load from the wheel in high-speed section in a railway system. Rail images in the Rail-5k dataset were captured by specialized cameras mounted on inspection cars riding along the railway, making the lens 200 mm vertically away from the rail surface and focusing vertically downward. We exclude images with shadows or overexposure on the rail surface for railway experts to label. We collected annotations for 1100 RGB images with $3648 \times 2736$ pixels in resolution, covering scenarios as tunnel, elevated bridge, straight and curve line, inner and outer rail, before and afer grinding or milling. fig. 1 shows the map of a typical rail section that we collect images. Each dot represents an image.

We also collected 3k images from uncurated images of rail surfaces. These images contains unknown corruption and unseen objects in the labeled set. fig. 3 shows some typical images in the unlabeled set.

In summary, our dataset contains two part of data, the first part is the labeled subset with a 1k labeled images, the second part is the unlabeled subset with 3k images. Thus our dataset can support both supervised and semi-supervised learning settings.

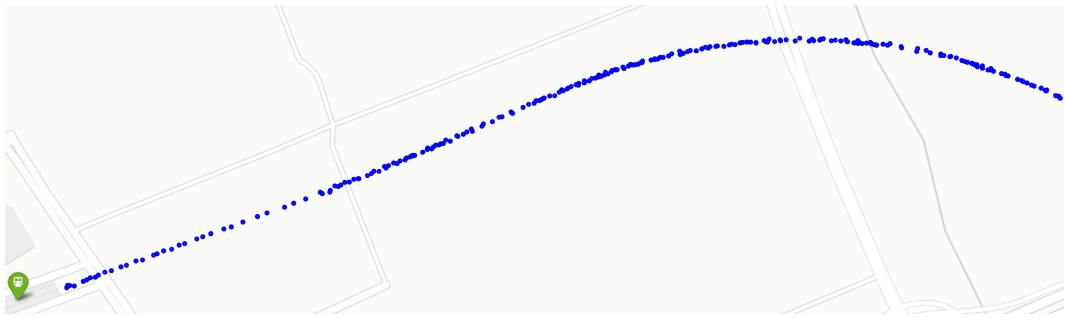

Figure 2: Map of typical sample points.

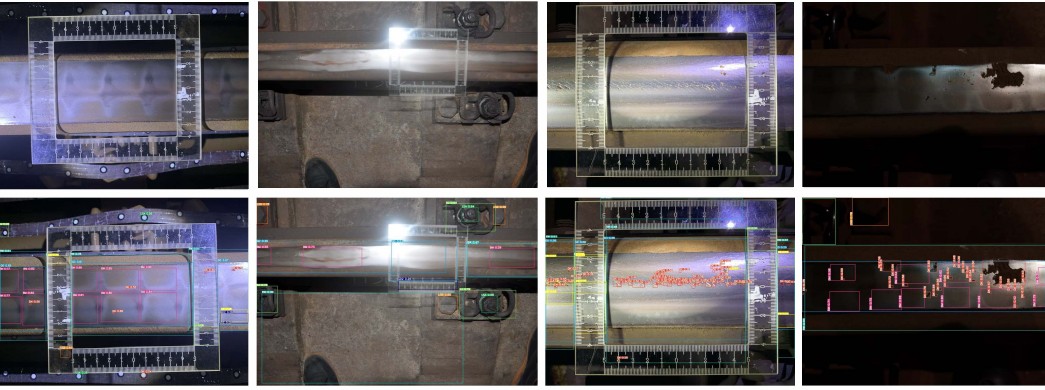

Figure 3: First line is corruption images, second line is prediction results. It can be observed that there are many false positives.

### 3.2 Fine-grained class definition and instance-level annotation

The annotations in our dataset were labeled by ten railway experts, each labeled images were at least checked by three experts. Based on the expert knowledge and railway standards, we use a fine-grained class definition and instance-level annotation paradigm for the railway defects detection. The labeling principle are listed in table 3.

Table 3: Annotation paradigm.

| Size | Boundary | Typical class | Annotation paradigm |
|---|---|---|---|
| Large | clear | Rail surface, Fastener, Screw | external rectangular box(same as common detection) |
| | obsure lump | Corrugation | wave valley of corrugation |
| Small | clear | Spalling,Indentation | stripped dent |
| Diffuse | sharp | Crack | union regions of small and dense boxes envelops cracking diffuse regions |

Note that the crack area are sharp and thin objects with no clear edge boundary, we annotate with a segmentation mask.

## 3.3 Dataset Splitting

We randomly split 20% of the 1,100 labeled images in to the test set, the remaining images are used as the training set in the supervised setting. For the semi-supervised setting, we use the same test set to evaluate the performance for comparison.

# 4 Annotation Statistics

In this section, we present the statistics of our dataset. The statistics are presented in three aspects, namely the image and bounding box distribution among class, the bounding box sizes and aspect ratios, and the center point of annotated bounding boxes.

## 4.1 Class distribution

fig. 4 shows the number of images and annotations containing each classes. The Burning and welded joint are ignored in our experiments and benchmark because of their rare appearance. The imbalance ratio with respect to the number of bounding box between the most majority class and the most minority is 40.98, the imbalance ratio with respect to the number of images is 6.07.

## 4.2 Sizes and aspect ratios of bounding boxes

(box size ratio graph) Bounding box annotations in our dataset vary dramatically in sizes and aspect ratios. There exist both tall and narrow objects as well as short and wide objects such as rail surface and contact band, normal square objects(fastener and screw). Besides, as shown in Figure 5, there are tremendous numbers of densely distributed small objects like spalling.

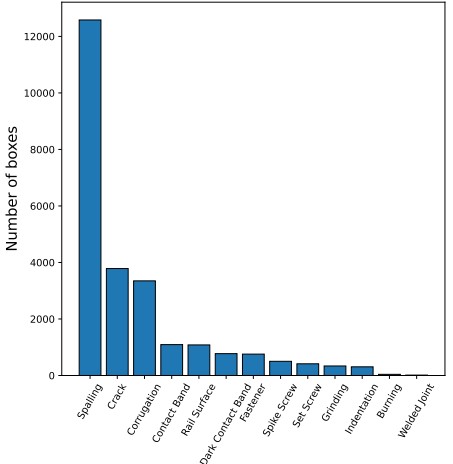

Figure 4: PR curve

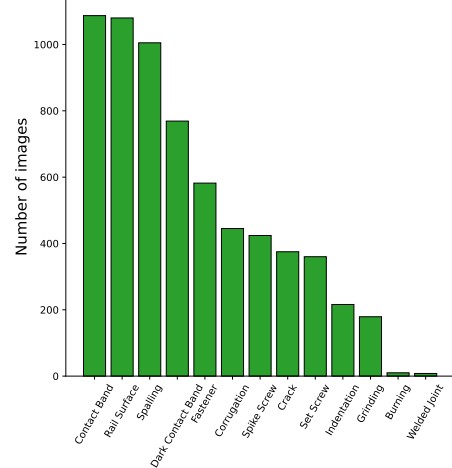

Figure 5: Concrete and Constructions

### 4.3 Object positions

Figure 7 shows the distribution of objects' center positions in our dataset. Because of the special shooting paradigm, rail surfaces usually lie horizontally or vertically in images. As a consequence, defects usually spread at the cross-zone in images.

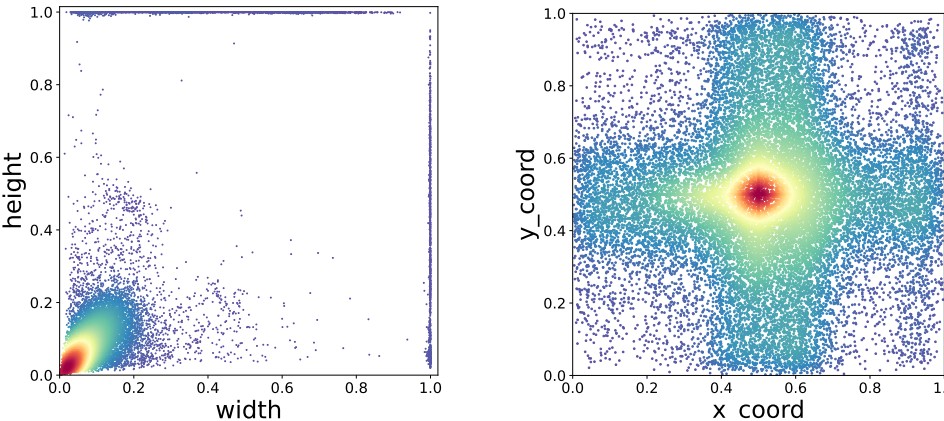

Figure 6: Width-height ratio of all annotations.    Figure 7: Center positions of all annotations.

## 5 Pilot Study on the Rail-5k Dataset

In this section, we conducted comprehensive experiments in several aspects to investigate the challenges and potential of the Rail-5k dataset. We trained an object detection model and a semantic segmentation model on Rail-5k as our baselines and showed the challenging attributes of our dataset.

Additionally, We proposed a semi-supervised benchmark for object detection.

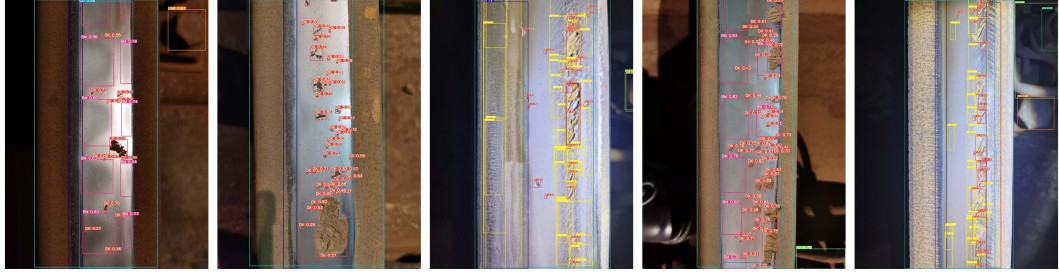

Figure 8: Typical prediction results on testset.

### 5.1 Benchmark for Detection

There are many popular detectors [20, 21, 15] on general object detection datasets. Recently, many new methods have been proposed and achieved the state-of-the-art results in the MS-COCO benchmark [16]. For example, YOLOv5 [13] is a light-weight model with mosaic augmentation and Generalized Intersection over Union(GIOU) loss. In our experiments, we finetined Yolov5-s as baseline on our dataset with MS-COCO pretraining. Detailed training settings are according to *data/hyp.finetune.yaml*[2]

It can be noticed that the detector's performance on crack is extremely low. This is because crack is more a texture than an object without clear definition of separated instances. Thus, we chose to tackle with this problem from another approach, which will be further discussed in Section 5.2.

---

[2]We implemented our experiments with Release v4.0 from https://github.com/ultralytics/yolov5/blob/develop/data/hyp.finetune.yaml

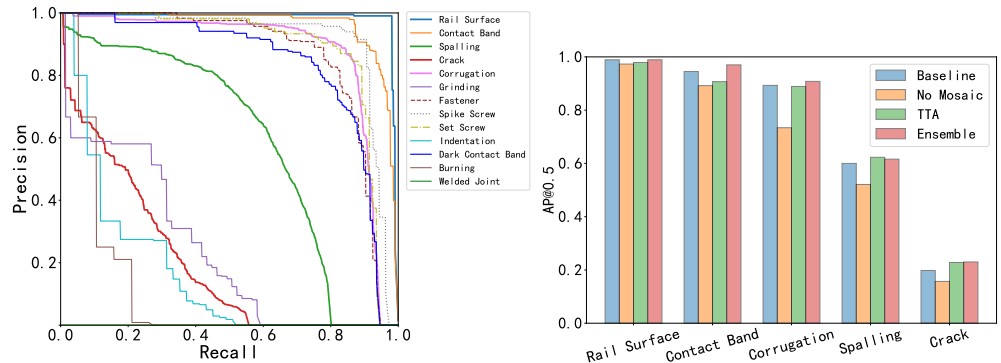

Figure 9: PR curve. | Figure 10: Ablation experiments.

Table 4: Metrics of baseline model for detection.

| Class | Precision | Recall | AP@0.5 | mAP@0.5:0.95 | AP |
|---|---|---|---|---|---|
| Rail Surface | 77.5 | 99.1 | 98.9 | 90.6 | 98.6 |
| Contact Band | 60.2 | 97.7 | 94.5 | 71.9 | 96.3 |
| Spalling | 33.2 | 74 | 60 | 24.8 | 58.9 |
| Corrugation | 60.3 | 91.2 | 89.3 | 48.2 | 87.6 |
| Grinding | 21.4 | 38.8 | 24 | 7.4 | 24.1 |
| Dark Contact Band | 64.4 | 81.4 | 76.7 | 36.7 | 83.4 |
| Fastener | 47.5 | 91.3 | 83.8 | 62.9 | 86.1 |
| Spike Screw | 37.8 | 92.5 | 86.8 | 48.6 | 91.8 |
| Set Screw | 58.6 | 88.2 | 87.3 | 52.2 | 88.5 |
| Indentation | 0 | 0 | 0.7 | 0.1 | 16.2 |
| Crack | - | - | - | - | - |

## 5.2 Benchmark for Crack

For cracking region, it is more of a texture and pattern than an object. Thus, we use segmentation to identify this class because the detection cannot recognize it well. We use Deeplabv3 [1] architecture with ResNet50 [9] backbone as segmentation model. The model is trained for 9000 iterations with a batch size of 16. We use SGD with momentum as the optimizer. Momentum and weight decay are set to 0.01, 1e-4 respectively. For the evaluation of our benchmark we choose the most common benchmark on segmentation, which is Intersection over Union(IoU). The model achieves 98.9% IoU on background and 67.8% IoU on crack, which is much better than the detection performance. DeepLabv3 can learn the main and obvious crack, but will ignore the tiny one.

Table 5: Metrics of baseline model for semi-supervised detection.

| Class | $s_{thr} = 0.6$ | $s_{thr} = 0.7$ | $s_{thr} = 0.8$ | $s_{thr} = 0.9$ |
|---|---|---|---|---|
| Rail Surface | 98.1 | 98.0 | 98.1 | 97.7 |
| Contact Band | 78.4 | 77.9 | 77.1 | 77.0 |
| Spalling | 60.1 | 58.9 | 57.9 | 58.2 |
| Corrugation | 89.6 | 89.2 | 89.5 | 88.6 |
| Grinding | 23.0 | 23.6 | 23.5 | 22.1 |
| Dark Contact Band | 92.7 | 92.9 | 93.1 | 92.4 |
| Fastener | 86.5 | 86.1 | 85.8 | 83.2 |
| Spike Screw | 93.2 | 94.6 | 91.3 | 87.4 |
| Set Screw | 88 | 88.5 | 87.2 | 85.4 |
| Indentation | 15.9 | 16.4 | 13.4 | 15.3 |
| Crack | - | - | - | - |
| mAP@0.5 | 63.29 | 63.27 | 62.43 | 61.55 |

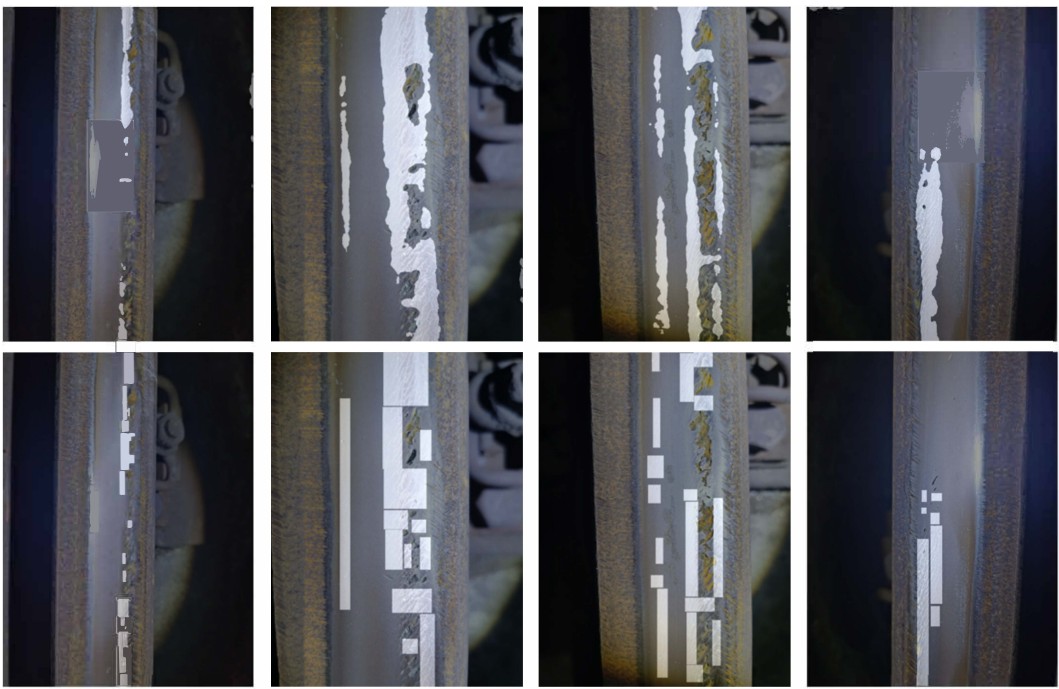

Figure 11: Images in first line are segmentation prediction results, and in second line are labels.

## 5.3 Benchmark for Semi-supervised Learning

With additonal 3k unlabeled images, we proposed a semi-supervised object detection benchmark. We presents results in table 5.

These results are generated with simple pseudo label technique. We inferenced on the unlabeled images with YoloV5-s trained following strategy described in Section 5.1. Then we apply a confidence score threshold$s_{thr}$ on all predictions and use remaining predictions as pseudo labels. Finally, we finetuned this model jointly on labeled images and unlabeled images with pseudo labels for 1 epoch and a base learning rate of 4e-4. Other training settings are the as ones in Section 5.1.

As shown table 5, detectors usually perform worse after being finetuned under semi-supervision. This could be caused by corruption and noise in unlabeled images.

## 6 Conclusion

We introduce Rail-5k, a real-world dataset for rail surface defects detection. We capture rail images across China and provide fine-grained instance-level annotations. This dataset poses new challenges both in rail maintenance and computer vision. As a baseline, we provide a pilot study on Rail-5k using off-the-shelf detection models. In later versions, Rail-5k will include more images and patterns, as well as more defects categories and image modalities, such as 3D-scan or eddy current data. This would make Rail-5k an even more standardized and inclusive real-world dataset. We hope this dataset will encourage more work on improving visual recognition methods for rail maintenance, particularly on object detection and semantic segmentation for real-world, fine-grained, small, and dense defects.

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
