# OpenReview forum: "Rail-5k: a Real-World Dataset for Rail Surface Defects Detection"
_NeurIPS.cc/2021/Track/Datasets_and_Benchmarks/Round1 — Submitted to NeurIPS 2021 Datasets and Benchmarks Track (Round 1)_

### Official Review · Reviewer_1PKH · 2021-06-30
**The introduction of the dataset is appreciated, but unfortunately several key concerns limit the contribution in their current state. It is recommended that the authors revise and resubmit their paper.**

**Rating:** 3
**Confidence:** 5

**Strengths:**

* The paper tackles a generally important and challenging problem. Our everyday’s infrastructure is an important element of our society and advancing our methods surrounding its maintenance should be of interest to many researchers.
* From what is visible in the example images and the corresponding bits of description, the dataset could be of interest even to the non civil engineering expert. From a machine learning standpoint the dataset can be of interest due to several defects varying significantly in size or frequency of occurrence.
* The appendix contains a dataset sheet

**Weaknesses:**

As much as I would like to see this published, particularly since availability of datasets surrounding infrastructure defects seems to seldomly be made available in open-access format, I unfortunately have multiple key concerns. There also is a general lack of clarity in writing.

I will raise some of the points here, and refer to the other weakness where appropriate in the sections on correctness, documentation, clarity and prior work.

Specifically:
* The writing and structure are poor. Figure and table captions are generally missing, and even worse, entire figures and tables remain unreferenced in the text. For instance, tables 1 and 2 are never once mentioned in the text. For example, in figure 2 it’s completely unclear what “large” “medium” and “small” actually translates to in practice.
* The main body of the paper would significantly benefit from a rewrite. On the one hand, the reader would benefit from a more detailed description. On the other hand, the existing text has a lot of redundancy. See clarity section. This is particularly unfortunate, given that the paper ends more than 1 page short of the allowed limit.
* It is unclear whether the dataset has been constructed in a meaningful way. Some figures, like figure 2 of the main body, suggest that more than 100 images were potentially taken in a short segment of the same rail track. Im also worried about the way the train and test sets were split. See correctness section.
* I understand the experiments are meant as a short showcase of what could be done, but I find them to be incomplete. First, there is no specification of how exactly section 5.1 has been trained. Second, when I observe the numbers in the table, I can see that some categories have tiny precision, such as 21, 31 or 37% for spalling, grinding, spike screw. This could be a consequence of many aspects, but it remains unclear to the reader what has been done. Is the performance for some classes so bad because there is very few data instances in the dataset? If so, have the authors done anything to address this imbalance during training? Third, the models are pre-trained on MS-COCO. This is a task unrelated dataset and it is unclear why this would be expected to help. As the authors point out themselves, the underlying problem is largely of texture nature.
* The authors list the additional 4k as a dataset contribution for investigation of e.g. corruption. I would appreciate a discussion on how the authors intend this dataset to be used. The suggested empirical showcase of table 5 does not look very appealing. In particular, what do the authors envision the researcher to do or investigate with this dataset?

**Additional Feedback:**

I truly believe that these type of datasets are important and certainly can be very meaningful to society. Unfortunately, in the present state of the paper, I do not believe the work to be ready for publication in NeurIPS yet. I would encourage the authors to revise and extend their main body, take a critical look at their conducted deep learning experiments and then attempt a resubmission.

**Clarity:**

Apart from some potentially contradictory statements (see previous sections) the paper should be rewritten to adhere to the expected NeurIPS standards. Specifically:
* The paper would benefit from an actual problem statement. I understand that many experts will be familiar with the types of shown defects, but in order for this to be a more general ML benchmark, readers will require a more in-depth problem statement and some sort of taxonomy of the investigated defects.
* Figure and table captions are mostly uninformative
* Tables 1 and 2 are unreferenced/unexplained in the main body.
* Figure 3 should have a better explanation
* Figure 8 is not referenced/explained
* Figures 9 and 10 are not referenced or discussed at all.
* As above points suggest, there is little to no discussion of results.

**Correctness:**

I am somewhat worried by the correctness of the paper because some of the statements made in the paper seem to be in contrast to each other.

* It is repeatedly stressed that 10 railway experts have annotated the data and then 2 additional experts conducted checks. However, in appendix lines 123+124 it is suddenly stated that “Images and videos in Rail-5k dataset are all collected by students and teachers in Tongji University, most of the images are collected and annotated by myself (Zihao Zhang). Without delving into the debate whether the students are experts or not (I'm sure they can be with sufficient training), this seems to suggest that the majority of the dataset was in fact collected and annotated by a single person?
* Appendix figure 3 suggests that the data comes from 10 sources of railway, with all but one source acquired on a short time window within the last 2 years. The statement that the data has been acquired over a time spam of 10+ years seems a little overclaimed and somewhat misleading. In addition only 3 tracks seem to be annotated (according to appendix figure 3). I imagine the images could be quite redundant? There really should be a discussion and clarification here.
* More importantly, when I look at the illustration of figure 2, it is suggested that the roughly 100 images in most tracks were actually acquired in a small region of a rail track. The authors say they employ a completely random train/test split of 80/20%. With the acquisition density hinted at in figure 2, this would mean that there is a very high overlap in train and test sets in terms of expected statistics. I am not sure this is meaningful. To the best of my knowledge, best practice would be to conduct a train/test split according to actually separate rail tracks, in order to assess generalization.
* Figure 3 (which should actually be called table), says that only 4 classes are annotated (crack, spalling, corrugation, burning)  and the rest is in the unsupervised 4k set. I am not sure what this implies for the experiments of the main body, where many more classes are shown? This should be clarified.
* It is unclear why there is so many different annotation principles used. It almost seems like each and every defect has been annotated in a different fashion (Table 3). When I look at figure 11, this causes concern. It almost seems like the semantic segmentation prediction (first row) is partially more accurate than the ground truth (bottom row). This is because some of the defects seem to have been labelled as filled rectangles, whereas they aren’t in reality. This would make evaluation tremendously difficult (and potentially wrong). If there has been trade-offs in favour of annotation time versus annotation accuracy, this should be mentioned and discussed explicitly. (Together with limitations and drawbacks that generally remain undiscussed)
* The checklist suggests that the authors have conducted a discussion on limitations. I don’t think this is the case.
* The checklist suggests that the authors have reported statistical deviations on experimental repetitions. What I can see in the tables is single numbers, which either suggests that the authors have only conducted a single training run, or have reported some sort of mean. In either way, this should be clarified, and ideally replaced with the actual statistical deviations. Particularly, in a dataset where reported performances range all the way from 20 to 100% precision, I imagine this could be quite volatile.

**Documentation:**

* The main contribution is a dataset, please actually link to it in the main paper.
* It seems that the dataset is currently hosted through obscure dropbox and google drive links. I believe the NeurIPS benchmarks and dataset track has opted to go for a single review process to prevent this from being necessary. The dataset should really be hosted in a way that promotes easy accessibility and is persistent.
* I understand the authors have also created a Zenodo record (which would be one of my suggestions for above point), however it is publicly unaccessible. This seems to limit the impact of the paper, given that the dataset is the main contribution.
* On the one hand, the authors say that they plan to release their dataset under a Creative Commons (CC) type license. While I am no expert on licensing, I believe this license usually implies and encourages sharing and distribution. At the same time, the Zenodo link, and the text in the appendix also suggest that the authors plan to control the dataset and only distribute the dataset manually to people who register per mail. At least to me, this seems counterintuitive and doesn’t correspond to what I personally would expect a CC type licensing to be (especially since it would allow someone who downloads to redistribute the dataset). I might however be wrong here. Maybe the authors should chose a different license instead.

**Ethics:**

I do not believe that there is any undiscussed (or discussed) ethical concerns.

**Relation To Prior Work:**

I am not an expert in the particular topic of railway defects. However, defect detection in infrastructure is a well pursued topic with many recent publications, including several reviews, as well as publications in top ML and vision conferences. In contrast to the other raised concerns, this seems to currently be a less dramatic point, but I would recommend the authors to expand upon this section.

Something that presently seems somewhat out of place is the related work section (and dataset table) on a few arbitrarily chosen object detection datasets like COCO, VOC or ImageNet. Although it is true that these datasets enjoy a lot of popularity, they seem rather unrelated to the investigated topic. If the authors want to review additional datasets, it would maybe make more sense to include related datasets in terms of defects in civil infrastructure. In addition to rail ways, this could be pipes, tunnels, bridges, pavement and plenty of other existing datasets that at least feature a very similar set of defects (such as cracks being almost omnipresent).

**Summary And Contributions:**

The paper introduces the Railway-5k dataset. The dataset is composed of 1100 labelled images of rail surface defects and an accompanied roughly 4000 further unlabelled images. While the former have been acquired in a generally controlled fashion, the latter’s acquisition seems to remain uncontrolled and thus feature various potential corruptions or other statistical deviations. Labelled images have been annotated into 13 types of rail defects, which are showcased in the context of deep neural network training defect detection in a small empirical study.

---

### Official Review · Reviewer_oiFk · 2021-07-02
**A relevant dataset. The manuscript and experiments need to be improved.**

**Rating:** 3
**Confidence:** 4

**Strengths:**

The authors consider an important problem in computer vision, which is the
robust detection and localization of defects in real-world industrial images.
The inspection of railroad tracks is a challenging task and well suited
for the evaluation of existing methods as well as the development of new methods
in the field of computer vision. The provided dataset images are valuable since for the research community it is often difficult to get access to data from real industrial applications.

**Weaknesses:**

The presentation of the dataset should be significantly improved. In its current form, the paper is not easy to follow due to numerous inconsistencies, wrong figure captions, entire experiments not being described in the text, etc. For details, see my comments in the section on "Clarity".

The evaluation protocol intended by the authors is not well defined. The proposed dataset is mainly designed for the evaluation of object detection methods that predict bounding boxes. In their first experiment, the authors test the performance of an object detector and report its performance on all defect categories except for one - the category "crack". They consider this defect separately by training a semantic segmentation model. This defect is also the only one in the dataset for which semantic segmentation labels are provided. This leaves the very important question of how the authors intend the research community to work with this dataset in the future. Should the "crack" defect always be excluded from the evaluation of object detection models and treated as a semantic segmentation problem?

**Additional Feedback:**

Why are the "burning" and the "welded" defects included in the dataset, but not included in the benchmark? If the number of samples is not sufficient for an evaluation of these defects, I suggest removing them from the dataset altogether.

Figure 7 suggests that most of the defects are actually in the center of the image. Why is that? From the qualitative images provided in the paper, I did not get this impression.


**Clarity:**


Several figures are never mentioned/explained in the text of the manuscript (e.g. Figure 1, Figure 9, Figure 10). In particular, it is not clear what the different bars in the ablation study in Figure 10 correspond to: What does the abbreviation TTA mean? What is the "Ensemble" experiment? The authors should detail how/why these experiments were conducted and discuss the results in the text.

Several images and tables in the manuscript need significant improvement. The captions of Figure 4 and 5 do not correspond to the content of the Figures For example Figure 4 shows dataset statistics but the caption is "PR Curve". The authors should explain what the two images on the right in Figure 1 display - what do the different colors of the bounding boxes correspond to? The detection results in Figure 3 and Figure 8 are almost impossible to recognize.

Several dataset statistics are not sufficiently detailed or are reported ambiguously. For example, from the paper, it is not clear how many "uncurated" images are provided in the new dataset. Sometimes the authors mention "4k" images (lines 9, 26), sometimes "3k" (lines 78, 82, 138). No statistics for the uncurated dataset are provided in Table 1. In line 63 the resolution of dataset images is given as 5x10^3 pixels, however, in Table 1 it says 10^4. The number of images in the CRRC dataset is simply reported as "more than 1000" in Table 1. The image resolution of three datasets in Table 1 is not reported.

**Correctness:**


The authors randomly split all 1100 labeled images into a train / validation / test set. I am concerned that constructing the splits this way can lead to a dataset bias in two ways:

1. For some defects in the dataset, only a very small number of samples are available. How do the authors ensure that these defects are not, by chance, all sorted into the training set?
2. The images are rather densely sampled from several track sections. This is also apparent in Figure 2. I believe that the three dataset splits should ideally be constructed from entirely different track sections. At least, it should be ensured that the sample points between the dataset splits should be sufficiently far apart. Otherwise, it is likely to evaluate on test images that are acquired right next to one of the training images. This strongly deviates from the real application, where test images always come from entirely new tracks.

I would like to hear from the authors how exactly the splits are constructed and how such a bias in the dataset was avoided.

**Documentation:**

It would be nice to include more dataset statistics about the different track sections. How many different track sections were used? How many images were acquired per track section?

Table 3 suggests that the authors make a difference between "small" "large" and "diffuse"/"medium" defects. What are the criteria used for these? Visualising the differences with an example would be beneficial for a better understanding. The same holds for the annotation policy for the different defects.


**Ethics:**

The authors claim that their work "will never do harm to society" (l. 226). I would like to encourage the authors to rethink this statement and shortly discuss the potential negative social impacts of automated industrial inspection systems in their manuscript.


**Relation To Prior Work:**

There exist numerous datasets for the task of defect detection in other
industrial scenarios, apart from rail surface inspection,
 that the authors might want to discuss in their related work section, for example:

1. Magnetic Tile Defect Dataset (Huang et al. - Surface Defect Saliency of Magnetic Tile)

2. The MVTec Anomaly Detection Dataset (Bergmann et al.): MVTec AD - A Comprehensive Real-World Dataset for Unsupervised Anomaly Detection (CVPR 2019).

3. Steel surface dataset:
A noise-robust method based on completed local binary patterns for hot-rolled steel strip surface defects
Song et al.


**Summary And Contributions:**

The authors present a new real-world computer vision dataset called "Rail-5k".
It contains 1100 images of rail surfaces with 13 different types of defects
annotated with bounding boxes. It further contains around 4000 uncurated
images without any labels that can be additionally used for training in a semi-supervised
setting. The authors argue that their dataset is particularly
challenging due to the class imbalance between the 13 different defects present in their dataset.
The authors further conduct a first experiment on their dataset with a common object detection
and semantic segmentation model.

---

### Official Review · Reviewer_8brR · 2021-07-05
**Rail-5k: a Real-World Dataset for Rail Surface Defects Detection**

**Rating:** 4
**Confidence:** 4
**Clarity:** The paper should be rewritten to adhe…

**Strengths:**

The paper poposes a dataset which contains over 5k high quality images from railways across China, and annotated 1100 images with the help from railway experts to identify the most common 13 types of rail defects. 1k+ labeled images for supervised training and 4k unlabeled images for unsupervised training.

**Weaknesses:**

The organization of the dataset should be significantly improved, and it is hard to understand the paper.

**Additional Feedback:**

The manner of releasing the dataset should be re-considered.

The paper should be carefully revised.

**Correctness:**

The submission is a dataset, and the paper also provide evaluation on the dataset. However, it is hard to guarantee the labeling accuracy.

**Documentation:**

none

**Ethics:**

I think that there is no etical concerns.

**Relation To Prior Work:**

The paper discusses the differences with previous contributions.

**Summary And Contributions:**

The paper poposes a dataset which contains over 5k high quality images from railways across China, and annotated 1100 images with the help from railway experts to identify the most common 13 types of rail defects. 1k+ labeled images for supervised training and 4k unlabeled images for unsupervised training.

---

### Author Response · Authors · 2021-07-14
**Withdraw, revise and resubmit to the next round**

We really appreciate the helpful feedbacks from reviewers. And we are going to revise the paper carefully and improve the writing, then we will resubmit to the second round. Thank you very much.

---

### Decision · Program_Chairs · 2021-07-26

**Decision:**

Reject

**Comment:**


The paper proposes a valuable dataset for the research community since, as reviewers mentioned, it is often difficult to get access to data from real industrial applications. However, in its current form, the paper needs several improvements to its clarity, documentation, evaluation protocol, and technical correctness. We encourage the authors to revise and resubmit the paper as planned.